# Asymmetric-flow field-flow fractionation of prions reveals a strain-specific continuum of quaternary structures with protease resistance developing at a hydrodynamic radius of 15 nm

Leonardo M. Cortez[1,2]*, Satish K. Nemani[1,2], Camilo Duque Velásquez[1,3], Aishwarya Sriraman[1,4], YongLiang Wang[1,4], Holger Wille[1,4], Debbie McKenzie[1,3], Valerie L. Sim[1,2]*

1 Centre for Prions and Protein Folding Diseases, Edmonton, Alberta, Canada, 2 Department of Medicine, Division of Neurology, University of Alberta, Edmonton, Alberta, Canada, 3 Department of Biological Sciences, University of Alberta, Edmonton, Alberta, Canada, 4 Department of Biochemistry, University of Alberta, Edmonton, Alberta, Canada

* lcortez@ualberta.ca (LMC); valerie.sim@ualberta.ca (VLS)

## Abstract

Prion diseases are transmissible neurodegenerative disorders that affect mammals, including humans. The central molecular event is the conversion of cellular prion glycoprotein, PrP$^C$, into a plethora of assemblies, PrP$^{Sc}$, associated with disease. Distinct phenotypes of disease led to the concept of prion strains, which are associated with distinct PrP$^{Sc}$ structures. However, the degree to which intra- and inter-strain PrP$^{Sc}$ heterogeneity contributes to disease pathogenesis remains unclear. Addressing this question requires the precise isolation and characterization of all PrP$^{Sc}$ subpopulations from the prion-infected brains. Until now, this has been challenging. We used asymmetric-flow field-flow fractionation (AF4) to isolate all PrP$^{Sc}$ subpopulations from brains of hamsters infected with three prion strains: Hyper (HY) and 263K, which produce almost identical phenotypes, and Drowsy (DY), a strain with a distinct presentation. In-line dynamic and multi-angle light scattering (DLS/ MALS) data provided accurate measurements of particle sizes and estimation of the shape and number of PrP$^{Sc}$ particles. We found that each strain had a continuum of PrP$^{Sc}$ assemblies, with strong correlation between PrP$^{Sc}$ quaternary structure and phenotype. HY and 263K were enriched with large, protease-resistant PrP$^{Sc}$ aggregates, whereas DY consisted primarily of smaller, more protease-sensitive aggregates. For all strains, a transition from protease-sensitive to protease-resistant PrP$^{Sc}$ took place at a hydrodynamic radius ($R_h$) of 15 nm and was accompanied by a change in glycosylation and seeding activity. Our results show that the combination of AF4 with in-line MALS/DLS is a powerful tool for analyzing PrP$^{Sc}$ subpopulations and demonstrate that while PrP$^{Sc}$ quaternary structure is a major contributor to PrP$^{Sc}$ structural heterogeneity, a fundamental change, likely in secondary/tertiary structure, prevents PrP$^{Sc}$ particles from maintaining proteinase K resistance below an $R_h$ of

**Data Availability Statement:** All relevant data are within the manuscript and its Supporting Information files.

**Funding:** This research was supported in part by Genome Canada in support of the Systems Biology and Molecular Ecology of Chronic Wasting Disease project (DM), the Alberta Prion Research Institute in collaboration with the Alberta Livestock and Meat Agency (Exploration IV, 201600016/2016A003R) (VS), the CJD Foundation (LC, VS), and the Alberta Synergies in Alzheimer's and Related Disorders (SynAD) program which is funded by the Alzheimer Society of Alberta and Northwest Territories through their Hope for Tomorrow program and the University Hospital Foundation (VS, LC). The funders had no role in study design, data collection and analysis, decision to publish, or preparation of the manuscript.

**Competing interests:** The authors have declared that no competing interests exist.

**Abbreviations:** AF4, asymmetric-flow field-flow fractionation; BH, brain homogenate; DLS, dynamic light scattering; DY, Drowsy prion strain; HY, Hyper prion strain; MALS, multiangle light scattering; NBH, normal brain homogenate; PK, endoproteinase K; PrP, prion protein; PrP$^C$, cellular form of prion protein; PrP$^{res}$, protease-resistant form of prion protein; PrP$^{Sc}$, disease-associated form of prion protein; PrP$^{sen}$, protease-sensitive form of prion protein; SHaPrP, recombinant Syrian hamster prion protein.

15 nm, regardless of strain. This results in two biochemically distinctive subpopulations, the proportion, seeding activity, and stability of which correlate with prion strain phenotype.

## Author summary

Prion diseases are neurodegenerative diseases that include bovine spongiform encephalopathy (BSE or mad cow disease) in cattle, chronic wasting disease (CWD) in cervids and Creutzfeldt-Jakob disease (CJD) in humans. These diseases are caused by self-propagated misfolding and aggregation of the naturally occurring prion protein. Variations in the structure of prion aggregates are associated with distinct disease phenotypes, but how this prion structural heterogeneity translates into clinical presentation has been difficult to determine, largely because it is technically difficult to isolate and characterize the full range of prion structures from prion-infected brain. Here, we overcame this challenge by using a versatile fractionation technique, one that is strikingly unexplored in neurodegenerative research, and present the most detailed description, to date, of strain-specific prion subpopulations. We found that prion quaternary structure was a major contributor to structural heterogeneity. We also discovered that all prion strains studied underwent a significant structural change resulting in two distinctive subpopulations whose proportions correlated with the strain phenotype. Our work provides new insights into the molecular basis of prion strain variation and is a proof of concept that can be applied to other protein misfolding neurodegenerative disorders.

## Introduction

Transmissible spongiform encephalopathies (TSEs), also known as prion diseases, comprise a group of lethal neurodegenerative disorders that affect several mammalian species including humans [1,2]. The central molecular event in TSEs is the misfolding and aggregation of the host-encoded prion glycoprotein, PrP$^C$, into a wide variety of assemblies associated with disease, collectively called PrP$^{Sc}$ [3,4]. During the course of the disease, a constellation of PrP$^{Sc}$ particles, ranging from small, soluble, protease-sensitive (PrP$^{sen}$) oligomers, to large, less soluble, partially protease-resistant (PrP$^{res}$) fibrils, progressively accumulate in the brain. Under controlled experimental transmission conditions, a specific disease phenotype, characterized by incubation period, clinical signs of disease, PrP$^{Sc}$ glycotype, and tissue tropism, can be reproduced in hosts with the same genetic background, leading to the operational concept of prion strains [2,5]. It is well established that different prion strains arise from different conformations of PrP$^{Sc}$, but within a given strain, there is also a diversity of PrP$^{Sc}$ structures that may further contribute to phenotype [6–12]. However, the molecular basis linking PrP$^{Sc}$ strain-specific structures and disease phenotypes remains elusive, in large part because isolating such structurally and biochemically diverse PrP$^{Sc}$ particles is a challenging task.

One approach to isolating PrP$^{Sc}$ particles has been sedimentation velocity. This technique has generated data that supports a strain-specific size distribution of PrP$^{Sc}$ particles and has provided evidence for an association between PrP$^{Sc}$ quaternary structure and its infectivity [13–17]. However, the limited resolution of this technique, differences in sample preparation and experimental conditions among laboratories, and the challenges of accurately assigning size values to isolated particles, have created inconsistent descriptions of the PrP$^{Sc}$ subpopulations contained in a prion isolate. A higher resolution technique, size exclusion chromatography (SEC), has been also used to investigate PrP$^{Sc}$ structural heterogeneity within prion strains

[18–21], but limitations related to protein-stationary phase interaction and fractionation range have restricted SEC to the study of the smallest PrP$^{Sc}$ oligomers present in a prion isolate, excluding from the analysis a major proportion of the largest and, potentially, most biologically relevant PrP$^{Sc}$ particles.

We have adopted asymmetric-flow field-flow fractionation (AF4) as an ideal technique for isolating and characterizing PrP$^{Sc}$ particles. AF4 is a flow-based separation technique widely used for the isolation and characterization of nanoparticles and polymers in the pharmaceutical industry, and is increasingly being applied to the characterization of biological macromolecules, protein complexes, extracellular vesicles, and viruses [22–24]. Two perpendicular flows, forward laminar flow and crossflow, are applied to particles, which are then separated based on their hydrodynamic properties. The technique allows for high resolution and reproducibility, and by adjusting crossflow gradients during the fractionation run, a very wide range of particle sizes can be separated, from a few nanometers to several micrometers [25]. Unlike SEC, the absence of a stationary phase greatly reduces the problems of sample-stationary phase interaction, which is particularly troublesome for "sticky" particles like large PrP$^{Sc}$ aggregates. It also allows for fractionation of complex mixtures such as brain homogenate. The use of in-line multi-angle and dynamic light scattering (MALS and DLS) detectors allows for precise measurement of the size and shape of the eluting particles.

Surprisingly, the use of AF4 technology to study neurodegenerative disease-related protein aggregates has been very limited to date [26–29]. Within the field of prion disease research, AF4-MALS-DLS was first used to determine the size of the most infectious prion particle [26]. In this seminal work, PrP$^{Sc}$ from 263K-infected Syrian hamster brains was first highly purified and digested with proteinase K, then disrupted by detergent, sonication and freeze-thaw before being fractionated and analyzed. Recently, we developed a method for using AF4 on brain homogenates without the need for initial purification or sonication, and with in-line DLS measurements, we were able to ascertain the size distribution of PrP$^{Sc}$ particles in the brains of RML-infected mice [29]. Now we have applied our isolation method and AF4-MALS-DLS analysis to the characterization of inter- and intra-strain differences of subpopulations of PrP$^{Sc}$ particles in the brains of Syrian golden hamsters infected with three different prion strains. All three strains share the same primary PrP structure; two strains (Hyper and 263K) were derived from different sources but produce almost identical phenotypes and share biochemical properties, the third (Drowsy) has a different phenotype and biochemical profile [6,10,30]. Our high-resolution fractionation and analysis provides the most detailed description, to date, of PrP$^{Sc}$ particles in the brain at terminal stage of disease. We demonstrate that quaternary structure is a major contributor to PrP$^{Sc}$ heterogeneity, but for all strains studied, a fundamental change, likely in secondary/tertiary structure, prevents PrP$^{Sc}$ particles from maintaining proteinase K resistance below an $R_h$ of 15 nm. These distinct populations of PrP$^{sen}$ and PrP$^{res}$ also have strain-specific changes in glycosylation pattern and templating activity.

## Results

### Solubilization step results in a high yield of soluble PrP aggregates for all strains, with no change in PK-resistance patterns compared with starting material

Our goal was to isolate PrP$^{Sc}$ aggregates, defined as all types of PrP aggregates that are specifically associated with prion infection, whether directly infectious themselves and resistant to proteinase K or not. To solubilize PrP from membranes while preserving PrP$^{Sc}$ assemblies close to their "natural" state, we incubated strains 263K, HY or DY brain homogenate (BH) with 2% dodecyl-β-maltoside and 2% sarkosyl sequentially, as optimized by Tixador *et al.* [14].

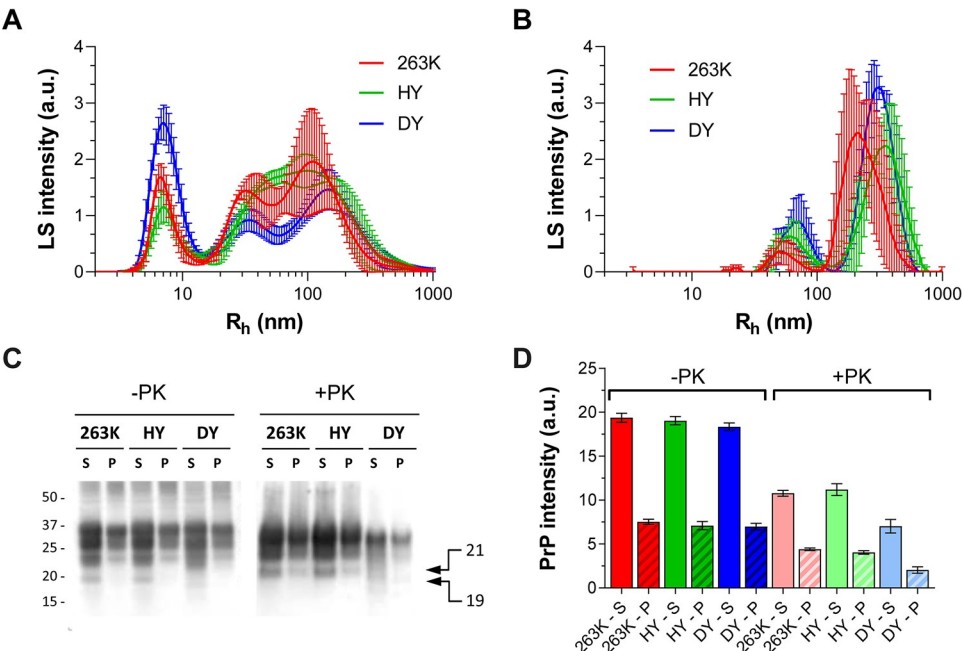

**Fig 1.** Particle size distributions determined by DLS measurements of detergent-soluble (**A**) and pellet (**B**) fractions after sequential incubation of 263K (red), HY (green), and DY (blue) brain homogenates with 2% dodecyl-β-maltoside and 2% sarkosyl, followed by 10 min centrifugation at 20,000 x g. Representative immunoblotting (**C**) and densitometric analysis (**D**) of detergent-soluble (S) and pellet (P) fractions of 263K (red), HY (green) and DY (blue). The pellet was resuspended in half of the original volume; the same volumes of S and resuspended P fractions were loaded in the polyacrylamide gel. Twice the volume of PK treated samples (20 μg/mL of PK for 1h at 37˚C) were loaded in the same gel for quantification. SAF83 (dilution 1:10,000) was used as primary antibody. Bars represent average ± SE of three BHs for each prion strain.

Detergent-insoluble particles that could affect the quality of our AF4 fractionations were removed by a short centrifugation. We performed batch mode DLS measurements on the supernatant and pellet and found similar sized particles were isolated for each strain: the supernatant contained particles of 3–300 nm $R_h$ compared with 40–600 nm $R_h$ in the pellet (Fig 1A and 1B). By immunoblot, the amount of PrP in the supernatant and pellet, before PK digestion, was similar for the three strains (Fig 1C and 1D), with a supernatant:pellet proportion of 72:28. Treatment of both supernatant and pellet with proteinase-K (PK) yielded the expected PrP$^{res}$ banding patterns, with the unglycosylated isoform migrating at 21 kDa for HY and 263K and 19 kDa for DY [30]. The ratio PrP$^{res}$:total PrP was higher for HY and 263K than for DY with values of 0.56 (±0.05), 0.59 (±0.04) and 0.38 (±0.08), respectively, for supernatant PrP and 0.58 (±0.05), 0.57 (±0.06) and 0.29 (±0.10) for the pelleted PrP. Assuming all animals have comparable levels of PrP$^C$, this was consistent with the fact that DY PrP$^{Sc}$ is less resistant to PK than 263K and HY PrP$^{Sc}$ [30].

## Strains with the same phenotype have the same PrP$^{Sc}$ size distribution

Batch mode DLS measurements of the soluble starting material indicated that particles had a size distribution of 3–300 nm $R_h$ (Fig 1A). We, therefore, optimized the AF4 crossflow gradient for separating particles of 3–300 nm $R_h$ within a 60 min run (see Methods for details). We injected 250 μg of total soluble protein and collected 200 μL fractions each minute. As expected, all fractograms had similar UV absorbance profiles (Fig 2A), indicating a similar

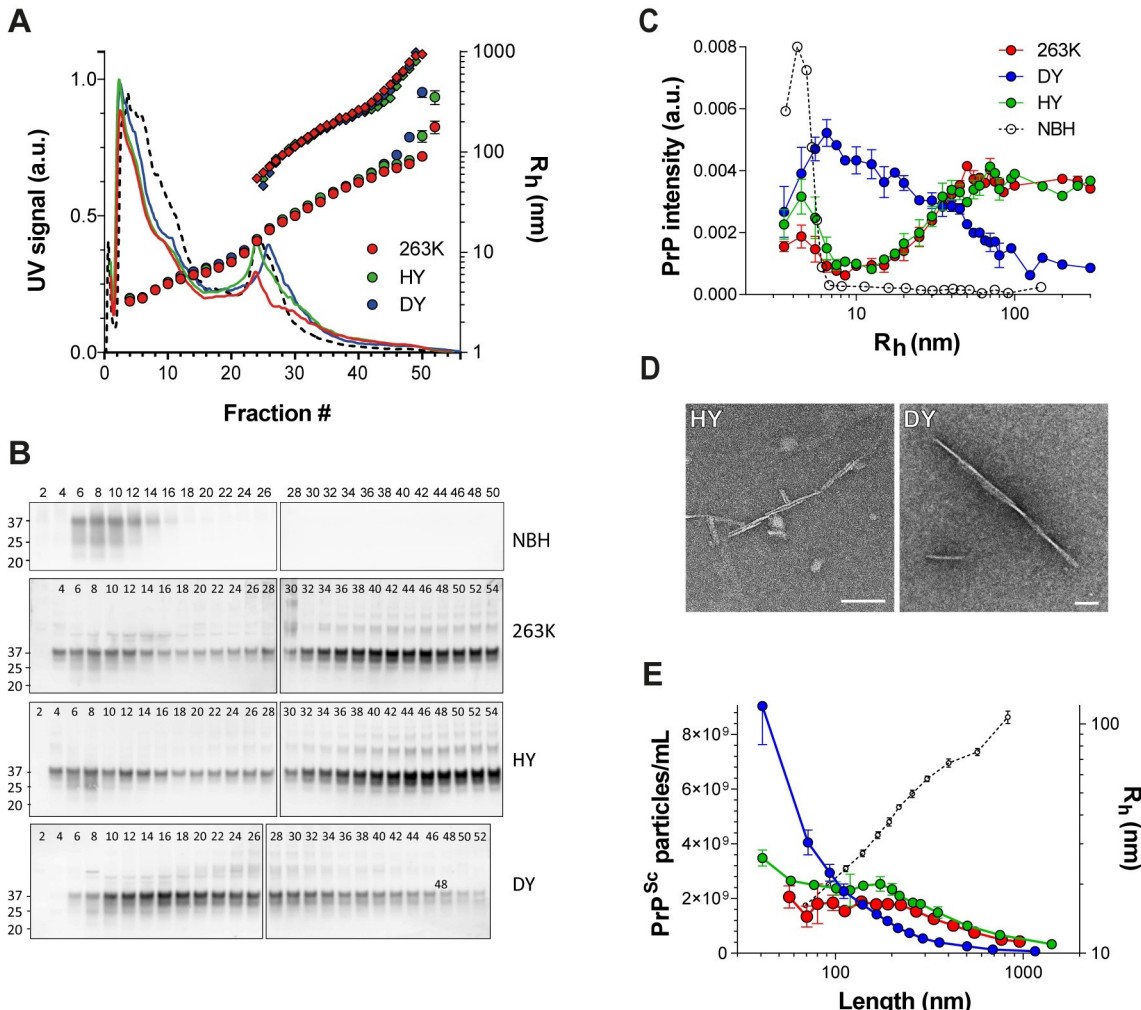

**Fig 2. A)** Representative fractograms of control (NBH, black dotted line), 263K (red), HY (green), and DY (blue) detergent-solubilized BHs. Solid curves: UV signal (protein content) at 280 nm; circles: $R_h$ (average ± SD) from thirty DLS measurements taken during the one-minute collection for each AF4 fraction; rhombus: particle length, calculated by fitting MALS signal to a rod-shape model. **B)** Representative immunoblots for PrP content in collected AF4 fractions; fractions from each brain homogenate were loaded in two polyacrylamide gels (gel 1: fraction 2 to 26/28, gel 2: fraction 28/30 to 50/54). SAF83 (dilution 1:10,000) was used as primary antibody. **C)** Size distributions of prion particles depicted as relative amount of PrP (from PrP immunoblot intensity; average ± SE) contributing to each particle size ($R_h$ (nm)). Non-infected (dotted line), 263K (red), HY (green), DY (blue). Three brains for each prion strain were analyzed and averaged. **D)** Electron micrographs of fractions #47 (> 70 nm $R_h$) from HY (left) and DY (right) AF4 fractionation. No fibrillar particles were visible on the same fraction number from NBH fractionation. Bars = 100 nm. **E)** Density of PrP$^{Sc}$ particles (average ± SE) for each particle length. 263K (red), HY (green), DY (blue). The correlation between particle length and $R_h$ is shown as a black dotted line.

amount and composition of proteins in the BH of the three strains. The $R_h$ of the eluting particles measured by in-line DLS matched that seen in batch mode DLS, 3–300 nm $R_h$.

To determine PrP elution profiles, fractions were immunoblotted for PrP and PrP signal intensity quantified (Fig 2B). The PrP intensity from each fraction was plotted against the average $R_h$ of the particles present in that fraction and the area under the curve from the entire run was normalized to 1 (Fig 2C).

Because our samples were not protease-digested before fractionation, the PrP in the sample contained normal PrP$^C$ in addition to PrP$^{Sc}$. To determine the contribution of PrP$^C$ to the

total PrP in the size distribution curves, we analyzed uninfected normal brain homogenate (NBH) under the same fractionation conditions (Fig 2B and 2C). All PrP$^C$ from age-matched uninfected brain homogenates eluted within the first 16 fractions, with the maximum PrP signal intensity corresponding to particles of ~4 nm $R_h$, and no signal in particles larger than 7 nm. Conversely, prion-infected samples had PrP particles ranging from 3–300 nm $R_h$.

Interestingly, almost identical PrP elution patterns were seen for 263K and HY, but there was a very different pattern for DY. For 263K and HY, the PrP signal intensity increased by four times from 7 to 40 nm $R_h$ particles, and the signal intensity remained almost constant as particle size increased beyond 40 nm $R_h$. In contrast, the PrP distribution for DY showed a constant decrease in PrP signal from 7 to 300 nm $R_h$. At ~30 nm $R_h$, the size distribution curves from the three strains intersected, indicating that DY contains more PrP$^{Sc}$ particles smaller than 30 nm, and 263K and HY have more PrP$^{Sc}$ particles larger than this.

## For PrP$^{Sc}$ particles increasing in size from 60–200 nm, particle density remains constant for 263K and HY but decreases for DY

To determine how many PrP$^{Sc}$ particles were present in each fraction (particle density), we needed information about particle length and PrP content for each fraction. Electron microscopy images of larger fractions from HY and DY indicated that both contain fibrillar material, likely prion fibrils based on appearance (Fig 2D). In addition, previous light scattering data on purified 263K particles has revealed that assemblies of 12 nm $R_h$ and larger are transitioning to a fibrillar form [26]. Therefore, we fit our MALS data to a rod-shaped model to determine particle lengths in fractions greater than 15 nm $R_h$ (Fig 2A and Table 1).

Once particle length was determined, we could use MALS data to directly calculate particle density (see Methods for details). However, because our starting material was brain homogenate, particles other than PrP$^{Sc}$ were present in the fractions. To overcome this issue, we analyzed only fractions 46–50, with particles of $R_h$ >70nm and highly enriched for PrP$^{Sc}$; these

**Table 1.  Particle density calculation.**

| Fraction # | $R_h$ (nm) | Length (nm) | PrP monomers per PrP$^{Sc}$ particle | PrP normalized intensity | Relative number of PrP$^{Sc}$ particles | Max. possible PrP$^{Sc}$ particle density (particles/mL) |
|---|---|---|---|---|---|---|
| 24 | 11.6 | 39.1 | 78 | 0.08316 | 0.001065 | 3.08E+09 |
| 26 | 15.5 | 57.8 | 116 | 0.102213 | 0.000884 | 2.56E+09 |
| 28 | 19.6 | 80.0 | 160 | 0.148777 | 0.000093 | 2.69E+09 |
| 30 | 22.9 | 106.6 | 213 | 0.179116 | 0.000084 | 2.43E+09 |
| 32 | 26.9 | 142.1 | 284 | 0.224708 | 0.000079 | 2.29E+09 |
| 34 | 32.9 | 175.1 | 350 | 0.348884 | 0.000996 | 2.88E+09 |
| 36 | 38.7 | 199.4 | 399 | 0.370763 | 0.000930 | 2.69E+09 |
| 38 | 46.0 | 223.5 | 447 | 0.322006 | 0.000720 | 2.09E+09 |
| 40 | 49.2 | 253.5 | 507 | 0.301998 | 0.000596 | 1.72E+09 |
| 42 | 57.6 | 361.2 | 722 | 0.290421 | 0.000402 | 1.16E+09 |
| 44 | 69.3 | 524.2 | 1048 | 0.299313 | 0.000286 | 8.26E+08 |
| 46 | 77.9 | 704.0 | 1408 | 0.309167 | 0.000220 | *6.36E+08* (*) |
| 48 | 105.6 | 1621.9 | 3244 | 0.394646 | 0.000122 | *3.52E+08* (*) |

The $R_h$ of particles was calculated from DLS measurements. The length of the particles was calculated fitting the MALS measurements to a rod-shape model. The number of PrP monomers per PrP$^{Sc}$ particle was estimated from the PrP$^{Sc}$ particle length, assuming a contribution of 0.49 nm in length per PrP monomer based on the recent parallel in-register intermolecular beta sheets (PIRIBS) model for the 263K strain. The PrP normalized intensity was determined by immunoblotting using Sha31 antibody. The relative number of PrP$^{Sc}$ particles was calculated by dividing the PrP normalized intensity by number of PrP monomers per PrP$^{Sc}$ particle. The maximum possible PrP$^{Sc}$ particle density was calculated with ASTRA software (see methods) for fractions 46 and 48, and extrapolated to the rest of the fractions based on the relative number of PrP$^{Sc}$ particles. (*) these values of particle density were calculated from MALS measurements directly, fitting the data to a rod-shape model, and assuming a rod radius of 4.25 nm. This calculation was performed for all samples; the specific values obtained for sample HY-E are shown in the table.

fractions had the highest ratio of PrP:total protein as determined by PrP immunoblotting intensity and UV absorption at 280 nm. In addition, the light scattering intensity and the corresponding particle density was much lower for NBH than for infected BHs in this region of the fractogram, suggesting that the main contribution to light scattering signal in fractions 46–50 was from large PrP$^{Sc}$ particles (S1 Fig). The PrP$^{Sc}$ particle densities for particles of R$_h$ ~ 80–90 nm were $4.51x10^8$ ($\pm 8.54x10^7$), $6.43x10^8$ ($\pm 1.43x10^8$), and $2.0x10^8$ ($\pm 1.61x10^7$) particles/mL for 263K, HY, and DY, respectively (Fig 2E and Table 1).

We also wanted to ascertain PrP$^{Sc}$ particle densities for smaller fractions. Having determined the particle lengths in each fraction, we then estimated the number of PrP monomers that would fit into fibrils of a given length, based on a parallel in-register intermolecular beta sheets (PIRIBS) architecture, with one PrP monomer present for every 0.49 nm in fibril height [31]. For particles of R$_h$ 15 nm, this equaled particles of length ~58 nm, comprised of 116 PrP monomers (see Table 1). Although lateral bundling of PrP$^{Sc}$ fibrils might also occur, its effect on mass per unit length calculations was not considered here.

Using the immunoblot data, we divided the PrP immunoblotting intensity from each fraction by the number of PrP monomers comprising the PrP$^{Sc}$ fibrils at that fraction. These values provided information about the relative amounts of PrP$^{Sc}$ particles in each fraction. Then we used the ratio of calculated particle density:PrP immunoblot intensity from fractions 46–50 to back-calculate the particle densities for the smaller fractions (Table 1).

PrP$^{Sc}$ size distribution plots indicated that HY and 263K had almost a constant PrP$^{Sc}$ particle density, $2.0x10^9$ particles/mL, in the range of 60–200 nm length (from ~15 to 40 nm R$_h$ according to DLS) (Fig 2E). Above 200 nm in length, the particle density decreased. DY PrP$^{Sc}$ showed a more pronounced and sustained decrease in the number of particles as length increased from ~50 to 800 nm.

## Glycosylation profile changes with the size of prion particle in a strain-specific manner

The ratio of di-, mono-, and un-glycosylated isoforms of PrP is one criterion for strain typing; however, 263K, HY and DY share the same PrP glycosylation pattern when unfractionated BH is analyzed by immunoblotting [6,32]. Our analysis of unfractionated BH showed the same di:mono:un-glycosylated ratio of 38:36:26 for total PrP (Figs 1A and 3A) and 55:35:10 for PrP$^{res}$ for all three strains. Given the differences in glycosylation ratios between total PrP and PrP$^{res}$, we questioned whether changes in glycosylation pattern might occur in different subpopulations of PrP$^{Sc}$ aggregates and, if so, whether these changes differed among strains. Indeed, immunoblot analysis of AF4 fractionations revealed strain-specific variation of PrP glycosylation patterns with the particle size.

The contribution of PrP$^C$ to the glycoform ratio in the first 16 fractions (particles smaller than 7 nm R$_h$) was determined by analyzing the NBH fractionation immunoblot (Fig 2B). More than 80% of PrP$^C$ eluted in fractions 6–10, where the glycoform ratio was 46:25:29 (Fig 3B). Between fractions 11 to 16, the percentage of di-glycosylated isoform increased to 75:10:15, as expected, given that extra glycan groups increase the R$_h$ of PrP$^C$. No appreciable PrP signal was detected after fraction 20, meaning that the contribution of PrP$^C$ to glycoform ratio calculations was limited to particles smaller than 7 nm R$_h$.

Given that the host PrP$^C$ is the same in all prion strains studied (Syrian golden hamster), we predicted that the glycoform ratios of PrP within fractions 6–10 would be similar across all strains. However, whereas HY and DY had almost the same glycoform ratios as NBH in these fractions, 263K had a higher proportion of di-glycosylated and less un-glycosylated isoforms (Fig 2B). This may indicate that there was less PrP$^C$ in these fractions, as suggested by the PrP

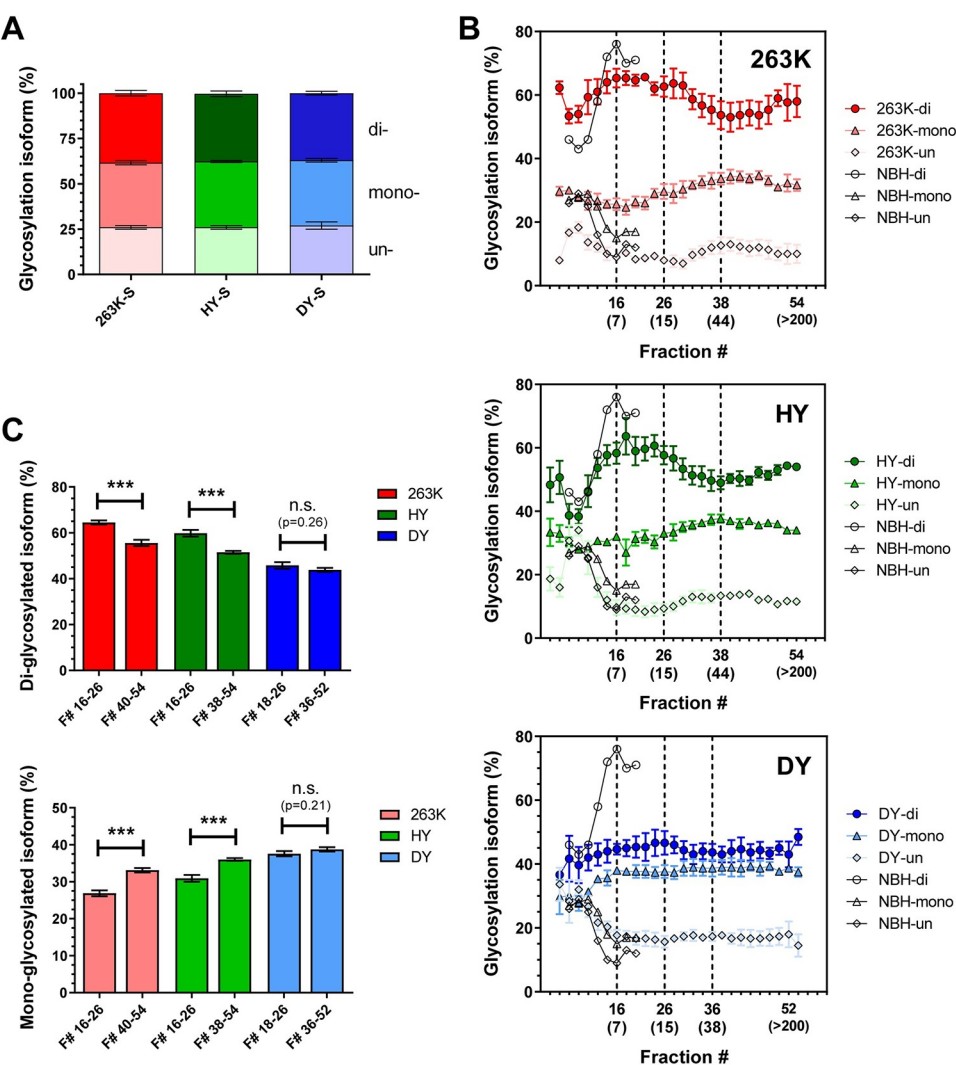

**Fig 3. Percentage of PrP glycosylation isoforms among AF4 fractions for 263K (red), HY (green) and DY (blue).**
**A)** Percentage of di-, mono-, and un-glycosylated PrP present in detergent-solubilized BHs as calculated from immunoblots of three BHs for each strain (representative immunoblot depicted in Fig 1C). **B)** Percentage of di- (circles), mono- (triangles), and un-glycosylated (rhombus) PrP isoforms present in each AF4 fraction as calculated from NBH (empty symbols), 263K (red), HY (green), and DY (blue) immunoblots of three BHs for each strain (representative immunoblot shown in Fig 2B). The $R_h$ (nm) of particles contained in fractions 16, 26, 36/38, and 52/54 are indicated in brackets. **C)** Percentage (average ± SE) of di- (top) and mono-glycosylated (bottom) PrP isoforms in the range of fractions where the glycoform ratio remains constant. Three BHs were analyzed for each prion strain. p-value *** ≤ 0.001.

size distribution analysis (Fig 2C), and/or a greater amount of small, highly di-glycosylated PrP^Sc oligomers contributing to the PrP signal, that shifted the glycoform ratio of this strain in these fractions.

For particles larger than 7nm $R_h$, 263K and HY had very similar glycosylation patterns. In particles of size range 7–15 nm $R_h$, ratios were 65:25:10 and 60:30:10 respectively; for particles 15–45 nm $R_h$, the ratio gradually shifted to 55:35:10 and 51:36:13 respectively, and then remained almost constant in particles of 45–300 nm $R_h$ (Fig 3B and 3C). Both the reductions in di-glycosylated and increases in mono-glycosylated isoforms were statistically significant for 263K and HY (Fig 3C).

Interestingly, DY had a distinct glycoform ratio which remained relatively constant at 45:38:17 for all particles larger than 7 nm. There was a trend towards a reduction in di-glycosylated particles with increasing size, but these changes were not statistically significant (Fig 3C).

## HY is comprised of more stable PrP$^{Sc}$ particles than 263K

PrP$^{Sc}$ structural stability is another feature used for prion strain typing. Incubation of brain homogenate with increasing concentrations of denaturant has previously demonstrated that 263K and HY are structurally more stable than DY [10,12,33,34]. Additionally, differences in the exposure of N-terminal regions in a narrow range of denaturant concentrations revealed a higher stability of HY over 263K at tertiary and secondary structure [10]. Instead of measuring the structural stability of the whole population of PrP$^{Sc}$ particles present in a BH, we explored the dissociating effect of SDS micelles over the different sub-populations of PrP$^{Sc}$ particles. Using the same fractionation conditions as before, we increased the concentration of sodium chloride from 20 to 150 mM in the AF4 running buffer, promoting SDS micellar formation (S2 Fig). Under SDS micellar conditions, all prion strains underwent a change in size distribution of PrP$^{Sc}$ particles (Figs 4A, 4C and S3). As expected, the proportion of particles with $R_h$ ~ 4 nm, corresponding to monomeric (and/or dimeric) PrP, significantly increased in all cases. For DY, the amount of monomerization was the largest, increasing from 10% to 50% of the

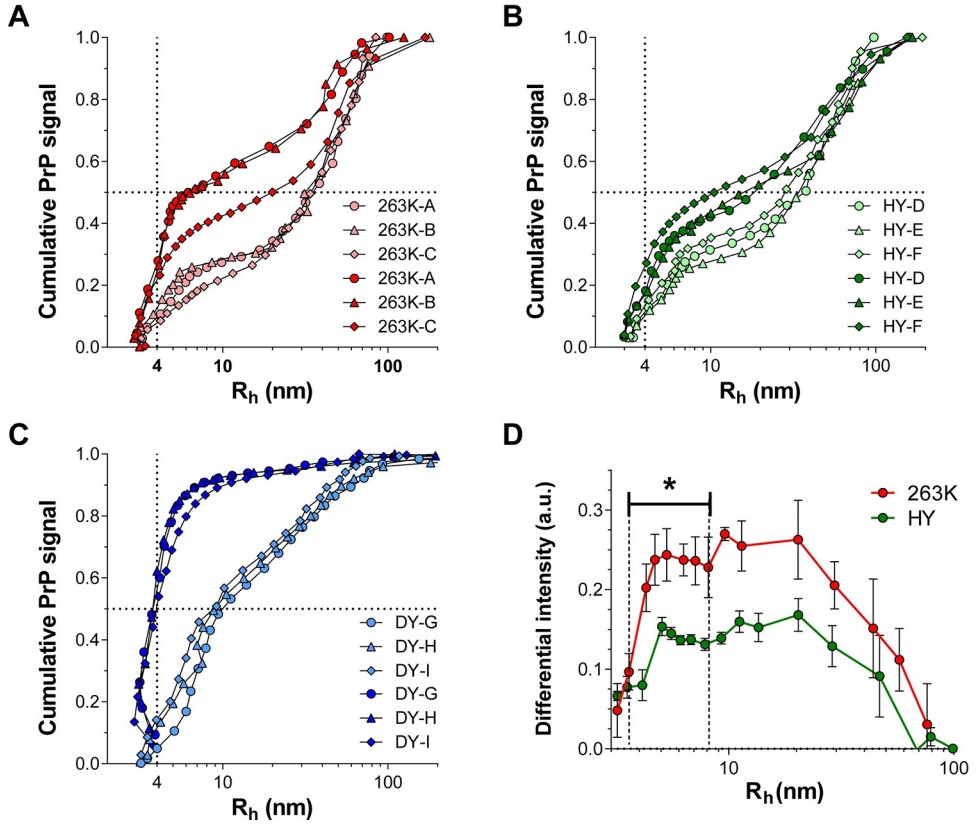

**Fig 4.** Cumulative PrP signal as a function of particle size for 263K (**A**), HY (**B**), and DY (**C**) fractionated in the absence (light symbols) or in the presence (dark symbols) of SDS micelles. Circles, triangles, and rhombus represent data obtained from three different BHs (indicated by the letters A-I). Vertical and horizontal dotted lines intersect at 4 nm $R_h$ and 50% of the cumulated PrP signal, respectively. **D)** Subtraction of the cumulative size distribution curves in the presence and absence of SDS micelles for 263K (red) and HY (green). Error bars represent SE; p-value $^* \leq 0.05$.

total population. In contrast, HY and 263K particles of $R_h \sim 4$ nm increased from 10% to only 25% of the population, indicating fewer particles were monomerized. Interestingly, AF4 fractionation also revealed subtle but statistical differences between 263K and HY. SDS micelles increased the population of particles with $R_h$ 4–10 nm more for 263K than for HY as measured by subtracting the cumulative curves from AF4 fractionation in the presence and absence of SDS micelles (Fig 4D). These results indicate a rank stability of PrP$^{Sc}$ quaternary structure of HY > 263K >>> DY.

## PrP$^{res}$ size distribution is independent of the strain phenotype

To investigate the association between PrP size and protease resistance, we PK-treated the AF4 fractions depicted in Fig 2B and quantified the amount of PrP$^{res}$ by immunoblotting (Fig 5A). We used Sha31, an antibody that recognizes the epitope 145–152 of the hamster PrP, which is within the PK-resistant core of PrP$^{Sc}$. The loss of epitope detection is interpreted as at least partial digestion of this epitope and the associated core. We therefore define PrP$^{res}$ as those particles whose Sha31 epitope remains stable under conditions of PK digestion. As expected, the AF4 fractions containing the smallest particles showed a lower PrP signal after PK digestion [13,15,18]. Surprisingly, despite the differences in size distribution of total PrP between HY, 263K and DY, immunoblotting analysis showed a very similar pattern of the PK resistant ~20 kDa fragment and its mono- and di-glycosylated isoforms, with a signal increase starting at fractions 26–28 in the three strains (Fig 5B). We plotted PrP$^{res}$ size distribution by correlating the $R_h$ measurements of AF4 fractionation, before PK digestion, with the immunoblotting intensity of the PK-treated fractions. PrP$^{res}$ signal appeared rather abruptly at ~15 nm $R_h$, and the intensity of PrP$^{res}$ signal reached a maximum at ~40–50 nm $R_h$, and was sustained until >200 nm $R_h$ (Fig 5C and 5D). This result strongly suggests that PrP$^{Sc}$ particles cannot maintain PK-resistance below 15 nm $R_h$, regardless of strain.

## Within each strain, PrP$^{res}$ particles have similar stability regardless of size

Next, we analyzed the structural stability of PrP$^{res}$ particles, defined here as the presence of epitope 145–152 under PK digestion conditions, as a function of size by incubating fractions larger than 25 nm $R_h$ with increasing concentrations of SDS followed by digestion with PK. As expected, immunoblotting analysis showed that PrP$^{res}$ particles were more stable for 263K than for DY (Fig 6). Under our experimental conditions, 263K PrP$^{res}$ intensities could be fitted to a sigmoidal curve, with 50% signal reached at 0.09–0.11% SDS. Conversely, DY PrP$^{res}$ signal was better fitted to a single exponential decay model with its 50% intensity occurring at 0.05–0.08% SDS (Fig 6). Regardless of inter-strains differences, the structural stability of PrP$^{res}$ particles within a given strain remained constant despite increasing size, at least in the range from 25 to 65 nm (Fig 6C). These results suggest that after acquiring PK resistance, no major conformational changes take place in PrP structure, at least around this epitope, other than the addition of more PrP units.

## PrP$^{Sc}$ particle seeding activity varies with particle size in a strain-specific manner

It has been previously demonstrated that HY has a higher templating activity than DY *in vitro* and *in vivo* [12,35,36]. Here we used RT-QuIC assay to compare the templating activity of the AF4 fractions containing PrP particles of 10, 20, 40 and 70 nm $R_h$ from 263K, HY and DY (Fig 7). For 263K and HY, the aggregation of recombinant hamster-prion protein (SHaPrP) had a shorter lag phase when seeded with fractions containing larger particles (40 and 70nm $R_h$) than when seeded with the same volume of fractions containing smaller particles (10 and 20nm $R_h$)

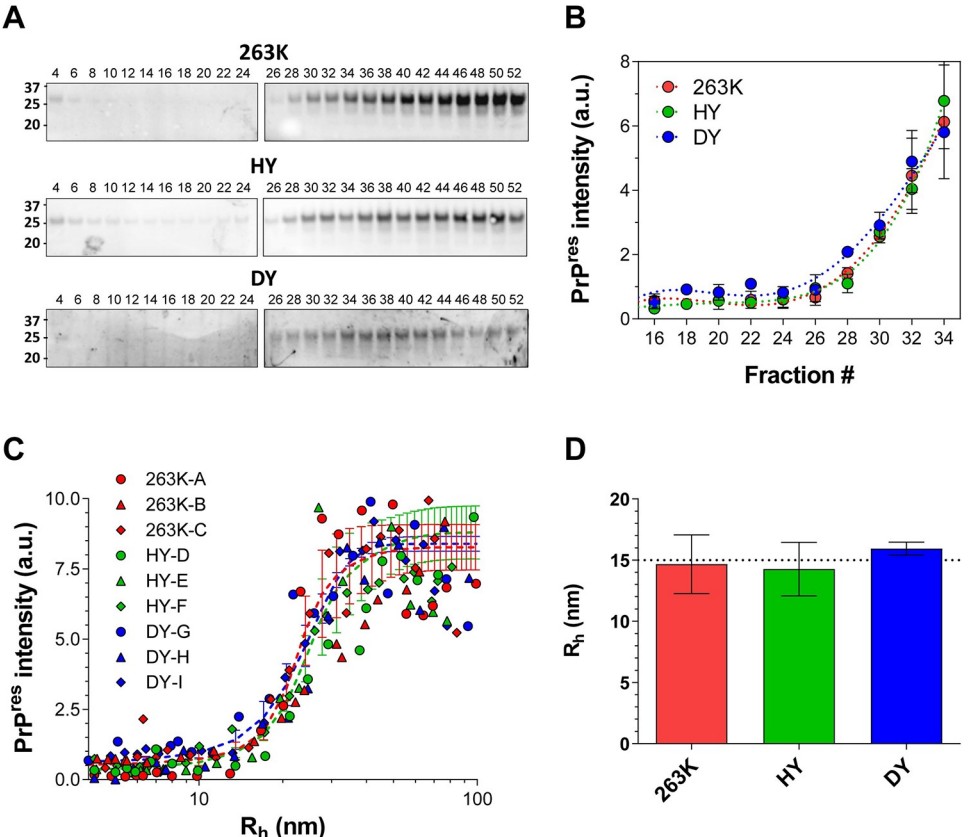

**Fig 5. A)** Representative immunoblot for PrP$^{res}$ from 263K, HY, and DY AF4 fractions; fractions from each brain homogenate sample were treated with PK and loaded in two polyacrylamide gels (gel 1: fraction 4 to 24, gel 2: fraction 26 to 52). **B)** Quantification of PrP$^{res}$ signal from 263K (red), HY (green), and DY (blue) AF4 fractions; (average ± SE). **C)** Size distributions of PrP$^{res}$ particles from 263K (red), HY (green) and DY (blue). Circles, triangles, and rhombus represent data from the three BH analyzed for each strain. Each set of data was fitted to a sigmoidal curve and the average of the curves is shown as a dashed line for each strain. **D)** Average size of the smallest PrP particle with increased PrP$^{res}$ signal. Error bars represent SE.

(Fig 7A, 7B and 7D). Interestingly, all tested DY fractions had the same seeding activity, with lag phases comparable to the slower HY and 263K fractions (Fig 7C and 7D). We then evaluated the ability of these fractions to convert PrP$^C$ into PrP$^{res}$ in CAD5-PrP−/− (HaPrP) cells, a murine catecholaminergic cell line lacking endogenous mouse PrP expression and expressing SHaPrP [37]. After seven passages of CAD5-PrP−/− (HaPrP) cells post-exposure to particles of 8, 10, 30 and 70 nm R$_h$ from HY, the highest PrP$^{res}$ signal was found in cells treated with 30 and 70 nm R$_h$ particles (Fig 7E). Interestingly, we also detected PrP$^{res}$ in cells treated with the 8 nm R$_h$ particles, which are PrP$^{sen}$ particles as assessed by Sha31 antibody. While there are no reports of successful infection of CAD5-PrP−/− (HaPrP) cells with DY, surprisingly, we were able to detect PrP$^{res}$ signal for cells treated with 30 and 70nm R$_h$ particles from DY, although with much less intensity than those from HY. The lack of infectivity of PrP$^{sen}$ DY particles, despite their intrinsic seeding activity, could be influenced by their lower structural stability, and consequent higher clearance, than the HY particles of the same size. To more directly compare the seeding activity of particles of the same size among prion strains, we seeded the RT-QuIC reaction with the same amount of total PrP monomeric units, as measured by immunoblotting. For

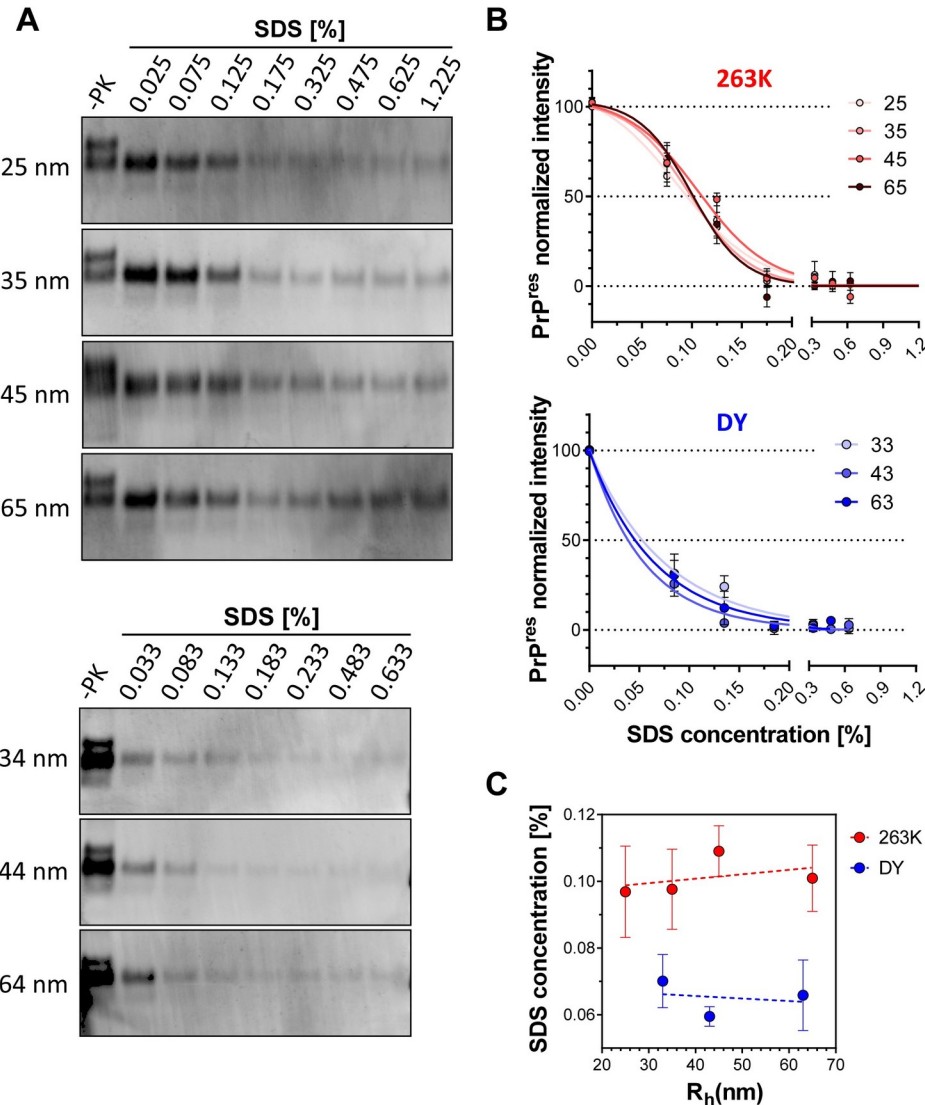

**Fig 6. Conformational stability of 263K and DY PrP$^{Sc}$ particles ranging from 25 to 65 nm R$_h$. A)** Representative immunoblots of 263K (top) and DY (bottom) fractions incubated with increasing concentrations of SDS and digested with PK. 3F4 (dilution 1:3,000) was used as primary antibody. **B)** Densitometric analysis of 263K (top) and DY (bottom) of remaining PrP$^{res}$ after incubation of AF4 fractions with increasing concentrations of SDS. 263K data was fitted to a sigmoidal function whereas DY data was fitted to a single decay hyperbolic function. Samples from three BH for each strain were analyzed. **C)** SDS concentration at which PrP$^{res}$ reaches half of the maximum intensity as a function of particle size. Data represents average of 3 replicates and error bars represent SE.

particles smaller than 20 nm R$_h$, HY and 263K showed higher seeding activity than DY, whereas for particles 40 nm R$_h$ and larger, DY particles were more efficient (Fig 7F). Of note, there was a statistically significant difference in lag phase between HY and 263K for particles bigger than 40 nm. For all these reactions, the amount of PrP added as seed was based on monomeric PrP content, not the number of PrP$^{Sc}$ particles. Thus, a direct correlation between seeding activity and size of PrP$^{Sc}$ was evident in the case of DY, with a shortening of lag phase as R$_h$ increases, but for HY and 236K, this correlation was harder to establish, since both templating activity and number of PrP$^{Sc}$ particles decreased with the increment of particle size.

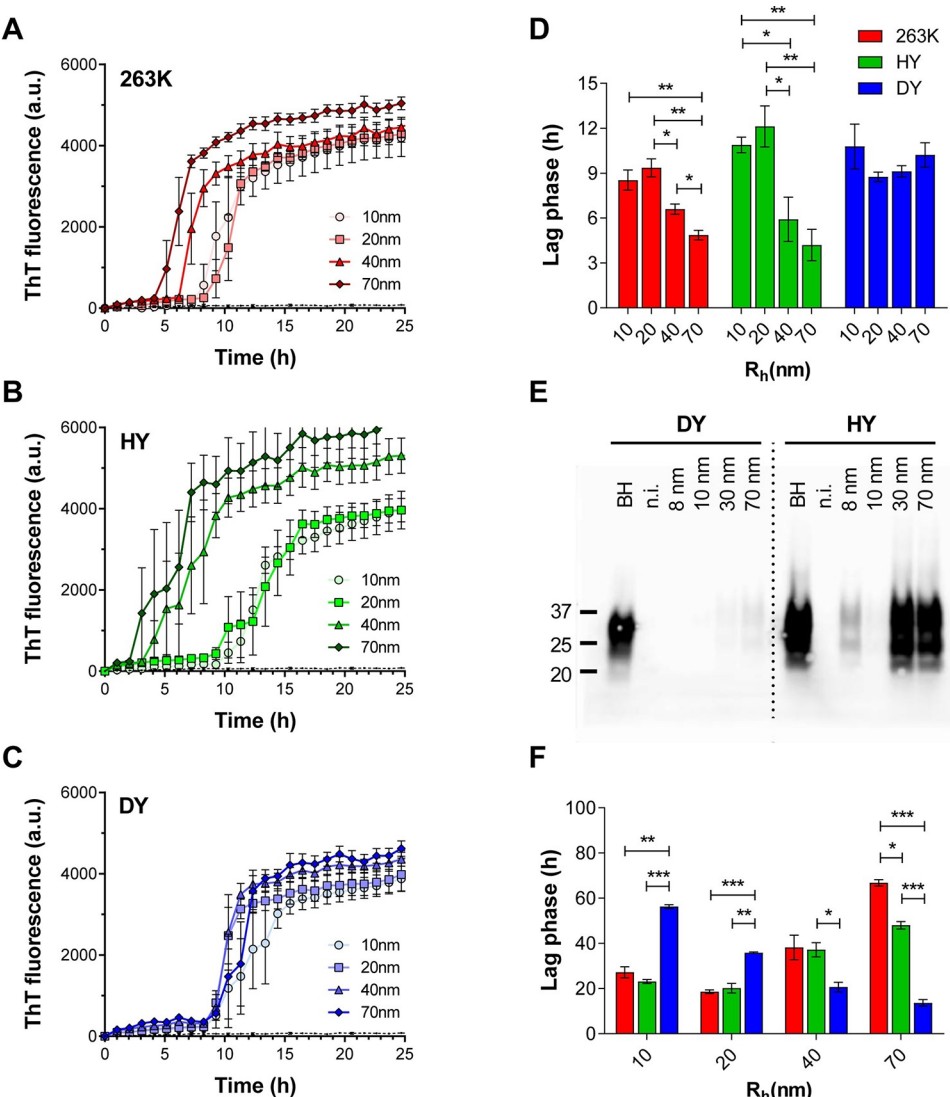

**Fig 7. Templating activity of 263K (red), HY (green), and DY (blue) fractions assessed by RT-QuIC assay and cell infectivity assay.** The kinetics of aggregation of recombinant SHaPrP unseeded (black dotted line) or seeded with particles of 10 (circles), 20 (squares), 40 (triangles) and 70 (rhombus) nm $R_h$ from 263K **(A)**, HY **(B)**, and DY **(C)** are depicted. All aggregation reactions were seeded with the same volume of AF4 fractions. **D)** Lag phases calculated from the kinetics depicted in A, B, and C. **E)** Immunoblot for PrP$^{res}$ accumulated in CAD5-PrP−/− (HaPrP) at passage #7 after incubation with AF4 fractions containing particles of 8, 10, 30 and 70 nm $R_h$. Brain homogenate and non-infected cells (n.i.) were used as positive and negative controls. **F)** Lag phases for the aggregation reactions of recombinant SHaPrP seeded with particles of 10, 20, 40 and 70 nm $R_h$, using the same amount of PrP mass. Data represents average of 3 replicates and error bars represent SE; p-value $^* \leq 0.05$, $^{**} \leq 0.01$, and $^{***} \leq 0.001$.

## Discussion

Prion strains consist of a spectrum of conformationally distinct PrP$^{Sc}$ particles that encode the information for a specific disease phenotype. However, the level of intra- and inter-strain heterogeneity of these PrP$^{Sc}$ particles, as well as the biological relevance of this heterogeneity, is poorly understood. Using an AF4-DLS-MALS configuration, we isolated and precisely determined the size distribution of PrP$^{Sc}$ particles in the brains of Syrian golden hamsters infected with 263K, HY or DY strains at end stage of disease. In addition, we provided a detailed

description of changes in glycoform profiles, resistance to protease, structural stability, and replication activity as a function of PrP$^{Sc}$ quaternary structure. Our data allow us to propose a model that explains PrP$^{Sc}$ heterogeneity.

## PrP$^{Sc}$ quaternary structure is associated with disease phenotype

Analysis of unfractionated PrP$^{Sc}$ from HY and 263K brain homogenate previously demonstrated that they have very similar secondary structures, structural stabilities, replication activities, PK digestion profiles, and glycosylation patterns [6,12,38,39]. Here we show that these two prion strains with almost identical phenotype also have almost identical size distributions of PrP$^{Sc}$ particles, further supporting the association between PrP$^{Sc}$ quaternary structure and strain phenotype. Conversely, DY, a strain with different clinical signs, incubation period, brain titre, and brain lesion profile, has a substantially different PrP$^{Sc}$ size distribution, with a larger population of small PrP$^{sen}$ particles and smaller population of large PrP$^{res}$ particles than HY and 263K. The PrP$^{sen}$:PrP$^{res}$ ratio was previously correlated with the incubation period in hamster-adapted prion strains, in which "fast" strains like 263K and HY have smaller PrP$^{sen}$:PrP$^{res}$ ratio than "slow" strains like DY [10,40]. Our results not only confirm this association between PrP$^{sen}$:PrP$^{res}$ ratio and incubation period, but also provide a detailed description of the size distribution and biochemical features of the PrP$^{Sc}$ particles comprising each of these two populations at end-stage of disease.

## The combination of particle density, structural stability, and replication activity of PrP$^{Sc}$ particles contributes to prion strain incubation period

In addition to the PrP$^{sen}$:PrP$^{res}$ ratio, PrP$^{Sc}$ chemical stability has been cited as an explanation for variable incubation periods. When unfractionated brain homogenate was treated with increasing concentrations of SDS or GdnHCl, an inverse correlation between PrP$^{Sc}$ chemical stability and incubation period was found for hamster-adapted prion strains [12,33,34]. Interestingly, the opposite situation was described for mouse-adapted prion strains, in which accumulation of less stable PrP$^{Sc}$ particles was found in strains with shorter incubation periods [41–43]. Two different hypotheses have been proposed to explain these converse correlations: a) the more stable hamster-PrP$^{Sc}$ particles have a lower rate of clearance in the brain, thus more time to replicate and to exert their neurotoxic effects, shortening the incubation period; b) the less stable mouse-PrP$^{Sc}$ particles have a higher rate of fragmentation which generates more replicative nuclei, causing accelerated PrP$^{Sc}$ propagation and shortening the incubation period [10,12,36,42–44]. Our study shows that both structural stability and templating activity change with the size of PrP$^{Sc}$ particles and do so in a strain-specific manner. Since the particle density of PrP$^{Sc}$ subpopulations is also strain-specific, we propose that the incubation period of a prion strain is a consequence of the combination of the stability, the templating activity, and particle density of each PrP$^{Sc}$ subpopulation accumulated in the brain. For example, highly replicating PrP$^{Sc}$ particles, if only present in low amounts, will not exert much effect on the overall replication activity of the strain. This is the case for DY. We found that PrP$^{res}$ particles from DY have higher replication activities than PrP$^{res}$ from HY and 263K, possibly because their lower stability creates more replication nuclei; however, the total amount of these DY PrP$^{res}$ particles is much lower than the amount of the less replicative HY and 263K PrP$^{res}$ particles. Furthermore, the predominant population in DY is PrP$^{sen}$, which has a lower replication activity than the PrP$^{sen}$ from HY and 263K. In addition, the lower stability of DY PrP$^{Sc}$ particles, which favours replication by creating new seeds, could also reduce replication capacity by promoting clearance. Thus, the combination of more PrP$^{sen}$ particles with low replication efficiency and structural stability and fewer, less stable PrP$^{res}$ particles with higher replication

efficiency, leads to an overall combined effect that DY is less efficient at replicating, resulting in a longer incubation period.

## Protease resistance develops at a hydrodynamic radius of 15 nm, regardless of strain

Surprisingly, despite the marked differences in size distribution of total PrP (PrP$^{sen}$ plus PrP$^{res}$) and in the ratio of PrP$^{sen}$/PrP$^{res}$ between the "fast" and "slow" strains studied here, the size distributions of PrP$^{res}$ are almost identical. In all cases, we observed a transition from PrP$^{sen}$ to PrP$^{res}$ starting at R$_h$ ~15nm. This result strongly suggests a strain-independent conformational change in PrP$^{Sc}$ structure once particles reach 15 nm R$_h$. Alternately, if the smallest particles with partial resistance to PK digestion simply came from the fragmentation of larger PrP$^{res}$ particles in the brain or during sample preparation, the 15 nm R$_h$ PrP$^{res}$ particles might not reflect the point at which conformational change occurs, but rather the smallest size at which a PK-resistant conformation can be maintained. Interestingly, we recently described the evolution of PrP$^{Sc}$ particles in the brain of RML-infected mice over the course of disease, observing that PrP$^{res}$ particles were also always 15 nm R$_h$ or larger [29].

To estimate how many PrP units would comprise a particle of 15 nm R$_h$, we made two assumptions, one assuming a fibrillar shape [26], and one assuming a parallel in-register intermolecular beta-sheet (PIRIBS) architectures [31]. This, combined with our MALS measurements, allowed us to estimate that fibrillar PrP particles of R$_h$ ~15nm will have a length of ~58 nm and consist of 116 PrP monomers giving a final mass of 3944 kDa, if we use an average M$_w$ = 34 kDa per monomer (based on the glycoform stoichiometry shown for 263K and HY, Fig 3). This size estimate is in line with other studies; sucrose gradient centrifugation and SEC studies of 263K and mouse-adapted scrapie identified PK-resistant particles starting at a M$_w$ higher than 2000 kDa [13,18]. We further validated our assumptions by comparing our M$_w$ calculation of larger particles to those reported by Silveira et al., on AF4-fractionated 263K. Silveira et al. found that 263K PrP$^{res}$ particles of R$_h$ ~37 nm are fibrillar particles with a calculated M$_W$ of 7770 (± 4270) kDa (by MALS) and corresponding to 362 (± 198) PrP units considering a PrP molecules averaging 21.5 kDa for these PK treated particles [26], comparable to our 39 nm R$_h$ particles with calculated 399 PrP monomers (Table 1).

We also did these calculations based on a four-rung β-solenoid structure containing two protofilaments, with two PrP monomers present for every 1.92 nm in fibril height, and a rod radius of 5.5 nm [45–48]. S1 table depicts the results for these calculations.

## AF4 fractionation reveals new strain differences

Although previous studies have not detected differences in glycosylation patterns between 263K, HY and DY [30,32], our data show that the transition from PrP$^{sen}$ to PrP$^{res}$ correlates with changes in glycosylation pattern and the degree of this change is strain-specific. Thus, we observed a more pronounced variation in glycotype stoichiometry for 263K and HY than for DY, which shows a lower and almost constant percentage of di-glycosylated isoform for all PrP$^{Sc}$ particles. Interestingly, lower levels of the di-glycosylated isoform have been associated with the accumulation of smaller and less stable PrP$^{Sc}$ particles in rodent prion strains [42,49,50]. Recent studies have demonstrated the relevance of PrP$^{Sc}$ glycotype stoichiometry in prion replication and faithful transmission of strain information [49,50]. Our data show that this stoichiometry either has an influence on, or is influenced by, the size of the PrP$^{Sc}$ assemblies.

AF4 fractionation also revealed subtle differences between the almost identical prion strains HY and 263K [33,38]. Using antibodies against the N-terminal region, Safar *et al.* reported a higher epitope exposure for 263K than for HY, when crude brain homogenates were incubated

with 2 to 3M GdnHCl [10]. Here we show that 263K PrP$^{Sc}$ particles dissociate in SDS at a higher rate than HY particles, demonstrating a difference in stability of PrP$^{Sc}$ assemblies between these two strains. In addition, RT-QuIC assays for the fractionated PrP$^{Sc}$ populations revealed a higher replication activity for HY particles with $R_h > 40$nm.

## A model for PrP$^{Sc}$ heterogeneity

Collectively, our data suggest that the main contributor to PrP$^{Sc}$ intra-strain heterogeneity is PrP$^{Sc}$ quaternary structure. PrP$^{Sc}$ particles grow in size with quasi-equivalent biochemical properties. Only one major change in the core structure (secondary/tertiary structure) takes place, and this change manifests as a switch from protease-sensitive to protease-resistant species, accompanied by a slight change in glycosylation stoichiometry, at a strain-independent size of 15 nm $R_h$ (Fig 8). Once PrP$^{Sc}$ becomes partially resistant to PK digestion, its conformational stability remains almost constant up to at least $R_h$ ~70 nm (or ~500 nm fibril length), suggesting that no other major conformational changes take place in PrP$^{Sc}$ core structure. This results in a continuum of quaternary structures, with the amount of PrP$^{Sc}$ particles comprising each size subpopulation, along with their specific stability and templating activity, defining the incubation period, and potentially the clinical phenotype, of a particular prion strain. Thus, we

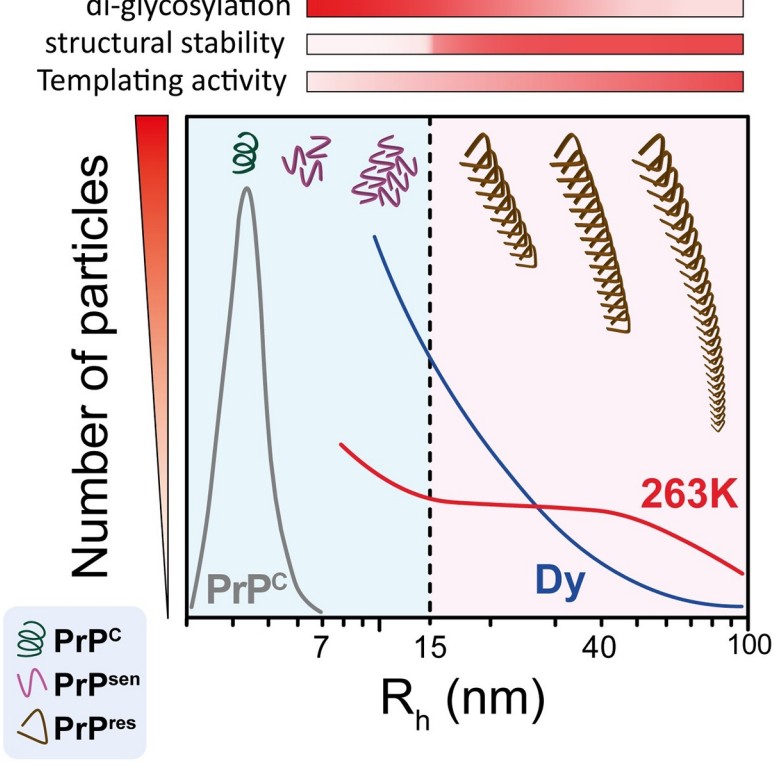

**Fig 8. Proposed model of PrP$^{Sc}$ structural heterogeneity.** A continuum of PrP$^{Sc}$ quaternary structures accumulate in the brain of prion-infected animals. Whereas the size of PrP$^{Sc}$ particles is the main contributor to PrP$^{Sc}$ heterogeneity, a major change in the core structure occurs when 15 nm $R_h$ is reached, generating two distinctive subpopulations that define the properties of the prion strain. At this size, there is a substantial increase in resistance to proteolysis and a reduction in the incorporation of di-glycosylated PrP isoforms into the PrP$^{Sc}$ particles. Once the mature PrP$^{res}$ particle is formed, the structural stability and glycosylation patterns remain constant despite further addition of PrP monomeric units. The PrP$^{Sc}$ templating activity is also affected by the size of the particle, increasing as the particle size increases, as seen in the case of DY.

could divide PrP$^{Sc}$ particles into different sized subpopulations and represent incubation period (IP) as follows:

$$1/IP \propto \sum_{k=1}^{n} \frac{PrP_k^{Sc} \; templating \; activity \times PrP_k^{Sc} \; nuclei \; generation}{PrP_k^{Sc} \; rate \; of \; clearance} \times PrP_k^{Sc} \; amount$$

where k is a PrP$^{Sc}$ subpopulation. The structural stability of PrP$^{Sc}$ particles will influence incubation period through 1) rate of clearance, and 2) rate of templating nuclei generation by fragmentation.

The above considerations regarding PrP$^{Sc}$ particle sizes assume that the isolated PrP$^{Sc}$ particles contain only prion protein, but it is likely that that other proteins and macromolecules could be interacting with PrP$^{Sc}$. If the degree of interaction varies with strain, this could affect the particle size distribution. Additionally, although we intentionally chose sample solubilization conditions that minimize the dissociation of PrP aggregates, the necessary presence of detergents in the solubilization and AF4 running buffers could still affect the actual size distribution of native PrP particles present in the prion infected brains, especially in the case of DY. Finally, although we divided PrP$^{Sc}$ particles into PrP$^{sen}$ and PrP$^{res}$ populations, this was based on Sha31 immunoblot detection, which is specifically probing for the resistance of epitope 145–152, so we cannot comment on the existence of assemblies with smaller resistant cores more C-terminal to this epitope.

## Conclusion

Our results show that PrP$^{Sc}$ quaternary structure is a significant contributor towards PrP$^{Sc}$ structural heterogeneity, resulting in biochemically distinctive PrP$^{Sc}$ subpopulations, and the proportion of these subpopulations correlates with prion strain phenotype. Our AF4-MALS approach to separating and analyzing prion aggregates from different prion strains has implications for other protein misfolding-related neurodegenerative disorders where conformational strains have been proposed [51–57]. The structural heterogeneity of α-synuclein, amyloid β, and tau aggregates could form the molecular basis for the broad spectrum of phenotypes seen in synucleinopathies, Alzheimer's disease, and other tauopathies. By applying our technique to these conditions, we may not only promote an understanding of strain-phenotype relationships across neurodegenerative diseases but help direct therapies towards the most relevant aggregate conformations.

## Methods

### Ethics statement

All animal studies described herein were performed in accordance with Canadian Council on Animal Care (CCAC) guidelines, under protocol AUP914 approved by the animal care use committee for Health Sciences Laboratory Animal Services at the University of Alberta.

### Hamster-adapted prion strains / brain tissues

Brain tissues were taken from clinically affected Syrian golden hamsters (Envigo) inoculated with brain homogenates (BH) of hamster-adapted strains of different origin: transmissible mink encephalopathy-derived Hyper (HY) and Drowsy (DY), and Scrapie-derived 263K. Three 263K-infected brains were obtained from hamsters inoculated with 10% 263K-stock BH and euthanized upon presenting with overt clinical signs at 74 days post infection (dpi). Seven HY-infected brains were generated by inoculation of 10% HY-stock BH, resulting in advanced clinical disease at 70 dpi. The DY samples included two brain homogenate pools; pool 1

consisted of three brains from hamsters inoculated with a $10^{-5}$ BH dilution of a DY-stock that were euthanized with clinical signs at 240 dpi. The second drowsy pool was composed of seven DY-infected brains from hamsters inoculated with a $10^{-5}$ dilution of the pool 1 and collected after >200 dpi. Two individual DY-infected brains from hamsters euthanized at 168 and 170 dpi were generated by inoculation of a $10^{-2}$ dilution pool 2 into naïve hamsters.

### Brain homogenization and sample blinding

Individual brains were weighed and homogenized in ultra-pure water (Sigma) with ceramic beads in an Omni Bead Ruptor to make 10% brain homogenates (w/v). The HY and DY pools were initially homogenized in ultrapure water using a dounce glass homogenizer. The resulting pool homogenate was further passaged through needles of different diameters, 17- 21G. All samples were aliquoted in sets of three for each strain (individual or pool homogenate) and assigned a sample number for blinded analysis. Aliquots were kept at -80˚C until further analysis.

### Batch mode dynamic light scattering (DLS)

DLS measurements of solubilized brain homogenate and pellets resuspended in solubilization buffer (50 mM HEPES pH 7.4; 300 mM NaCl; 10 mM EDTA; 4% (w/v) dodecyl-β-D-maltoside (Sigma)) were performed with a Malvern Zetasizer-Nano S. A 633 nm wavelength HeNe laser was used to detect backscattered light at a fixed angle of 173˚. Measurements were performed at 20˚C and the solution viscosity and refractive index of water was assumed for calculation purposes. The data was collected without attenuation and a minimum number of 10 consecutive runs of 10 seconds each was averaged to obtain the autocorrelation function. Particle size was calculated by the manufacturer's software through the Stokes-Einstein equation assuming spherical shapes of the particles.

### Asymmetric-flow field-flow fractionation (AF4)

Brain homogenates (10% w/v) were solubilized by adding an equal volume of solubilization buffer (50 mM HEPES pH 7.4; 300 mM NaCl; 10 mM EDTA; 4% (w/v) dodecyl-β-D-maltoside (Sigma)) and incubated for 45 min on ice. Sarkosyl (N-lauryl sarcosine; Fluka) was added to a final concentration of 2% and incubated on ice for 30 min. The sample was centrifuged (20,000xg, 10 min) at 4˚C. The supernatant was collected and 350 µg total protein were subjected to asymmetrical-flow field-flow fractionation on an AF2000 Postnova system using 50 mM HEPES pH 7.4 (containing 50 mM sodium chloride and 0.05% sodium dodecyl sulfate (SDS) to minimize interaction of proteins with the channel membrane) as the running buffer. The channel was 26.5 cm in length and 350 µm in height, constructed with a trapezoidal spacer of maximal width 21 mm at the inlet, and lined with a 10 kDa cutoff polyethersulfone membrane at the accumulation wall. Samples were focused for 4 min and then eluted at a channel flow of 0.5 mL/min with constant cross-flow for the first 10 min, decreasing from 1.5 to 0.35 mL/min in the following 15 min, from 3.5 to 0 mL/min in the next 30 min, and run with no cross-flow for the last 10 min. A slot pump was run at 0.3 mL/min to concentrate the samples before they passed through the detectors. Fractions of 0.2 mL were collected. Multi-angle light scattering (MALS) and dynamic light scattering (DLS) were collected simultaneously in the in-line DAWN HELEOS II detector (Wyatt Technology), operating at a wavelength of 662 nm. Data analysis was performed with ASTRA 6.1.7.17 software (Wyatt Technology).

### Hydrodynamic radius measurements from in-line DLS

DLS data were collected every 2 seconds at 140.1˚ scattering angle for determination of hydrodynamic radius ($R_h$) of the AF4 eluting particles. Fitting the data to an autocorrelation

function was performed using the cumulants model. The injection of 350 µg of protein in each AF4 run ensures sufficient sample to scatter at least three times more light than the solvent, providing good signal/noise ratios for all size ranges.

## Calculation of particle length using in-line MALS

Static light scattering data were collected at 16 detector angles simultaneously every 2 seconds across the fractogram. For particles with sizes above the angular variation detection limit ($R_g$ ~10 nm), we used the MALS data to determine particle length assuming a rod-shaped particle. This assumption is supported by previous AF4-MALS-DLS and electron microscopy analysis of highly purified 263K strain $PrP^{Sc}$ particles where it was shown that $PrP^{Sc}$ particles became ellipsoid-spherical at $R_h$ = 12.4 nm, and larger particles had a more fibrillary structure [26]. We also detected fibrillar structures in our larger fractions where PrP was enriched (Fig 2D). The excess Rayleigh ratio, $R(\theta)$, as a function of scattering angle, was fit to the Rayleigh-Gans approximation for an infinitely thin rod using the ASTRA software, in which the form factor, $P(\theta)$, obeys the following relationship [58]:

$$P(\theta) = \left(\frac{1}{u}\right) \int_0^{2u} \frac{\sin t}{t} dt - \frac{\sin^2 u}{u^2}$$

where u = [($\pi$ $n_o$/$\lambda_o$) L sin($\theta$/2)], L is the rod length and is assumed to be much greater than the rod diameter, $n_o$ is the refractive index of the solvent at the incident radiation (vacuum) wavelength, and $\lambda_o$ is the incident radiation (vacuum) wavelength.

## Number of PrP monomers per particle

Knowing particle size allowed us to calculate the number of PrP monomers that would "fit" into fibrils of different lengths. Based on the recently published model for 263K $PrP^{Sc}$ fibrils [31], we assumed that each monomer contributes with a height of 0.49 nm, giving the following calculation:

$$PrP\ monomers = \frac{fibril\ length\ (nm)}{0.49\left(\frac{nm}{monomer}\right)}$$

## Calculation of $PrP^{Sc}$ particle density

Based on the number of monomers present in each sized particle, we could use our immuno-blot data of PrP intensities (representing relative amount of total PrP monomers in each fraction) to further calculate the relative number of $PrP^{Sc}$ particles present in each fraction. The absolute number of $PrP^{Sc}$ particles present in each fraction could not be directly measured because there were proteins other than $PrP^{Sc}$ particles present; these other proteins contributed to the MALS signal that would normally be used to calculate particle density. However, we were able to determine the maximum number of $PrP^{Sc}$ particles/mL that could be present, by analyzing MALS data from fractions 46–50. These fractions have much less protein, as measured by UV signal, but have the highest level of PrP, based on immunoblot, making them the purest PrP fraction. By assuming all MALS signal came from $PrP^{Sc}$ particles in these fractions, we calculated the theoretical maximum number of $PrP^{Sc}$ particles/mL ($PrP^{Sc}$ particle density) present in these fractions. To determine the theoretical maximum $PrP^{Sc}$ particle density for the smaller fractions, we took the ratio of $PrP^{Sc}$ particle density:relative number of $PrP^{Sc}$ particles from fractions 46–50 and applied it to the smaller fractions (see Table 1 for details).

For particles of uniform density and volume $V$, the number of particles per mL in the $i^{th}$ slice, $n_i$, is proportional to the zero-angle Rayleigh ratio divided by the square of the particle´s

volume:

$$n_i \propto R(0)/V_i^2$$

The value of $R(0)$ was determined by fitting the light scattering intensity as a function of angle to the rod model. The refractive index of the prion particles was set to 1.587, as this is a typical average value for protein refractive index [59]. For estimation of particle volume, we used the particle lengths calculated from MALS data and assumed a rod radius of 4.25 nm [31]. From this, the ASTRA software (Wyatt Technology) was able to calculate the number of PrP$^{Sc}$ particles per mL for each eluting data slice across the fractogram [60].

## Proteinase-K digestion

For all Proteinase-K (PK) digestions, 0.5 μg/μL of protein was incubated with 20 μg/mL of PK at 37˚C for 1h and 300 rpm orbital shaking. For digestion of AF4 fractions, 10 μL of each fraction were incubated with 20 μg/mL of PK in the presence of 0.5 μg/μL bovine serum albumin (BSA) as the PK substrate to account for the low mass of total protein in eluted fractions. The reaction was terminated by addition of Pefabloc (Sigma) and incubation at 4˚C for 10 min, followed by boiling with SDS-sample buffer and electrophoresis on 4–12% NuPAGE Bis-Tris gels (Invitrogen).

## Conformational stability assay

The conformational stability assay was performed as described previously with slight modifications [12]. Briefly, 10 and 20 μL from 263K and DY AF4 fractions were incubated for 30 min. with increasing concentration of SDS, from 0.025 to 0.635% (w/v), at 22˚C and shaken at 450 rpm. The samples were then digested with 1 μg/mL of PK for 1 h at 37˚C and 450 rpm orbital shaking in the presence of 15 μg of bovine serum. The reaction was terminated by the addition of 4 mM PMSF. For 263K samples, equal volumes of 2x loading buffer (125 mM Tris HCl pH 6.8, 20% glycerol, 4 mM EDTA, 6%SDS, and 10% 2-mercapto ethanol) was added, boiled for 10 min., loaded in a 4–12% NuPAGE Bis-Tris precast gels, transferred to PVDF membranes, and probed with the 3F4 antibody (dil. 1:3,000; Sigma). To improve immunoblot PrP$^{res}$ signal, DY samples were methanol precipitated by adding 5x volumes chilled methanol and incubated for 2h at -30˚C. The samples were centrifuged at 18,200 x g for 30 min., and the pellets were dried, resuspended in 1X loading buffer, boiled for 10 min. and subjected to SDS-PAGE and immunoblotting with 3F4 (dil. 1:3,000; Sigma).

PrP signal was analyzed by Image Quant software and PrP$^{res}$ intensity was fitted to sigmoidal and one phase decay models with Graph Pad Prism 8.0.

## Expression and Purification of recombinant hamster PrP

Syrian hamster prion protein (SHaPrP) containing residues 90–231 (which form the protease-resistant fragment in PrP$^{Sc}$) was cloned into the pET-15b plasmid between the XhoI and EcoRI sites as described previously [61]. Cys residues were introduced at each terminus by mutating Ser residues in the thrombin cleavage site and the prion site S232. The 19-kDa, N-terminal His-tagged SHaPrP was expressed in *Escherichia coli* BL21 (DE3) and purified by FPLC (GE Healthcare) using a nickel-nitriloacetic acid (Ni-NTA column). PrP was refolded on the Ni-NTA column, with native folding confirmed by circular dichroism spectroscopy, and the purity and identity of the protein verified by SDS–PAGE and immunoblotting (3F4 antibody, Millipore).

## Real-time quaking-induced conversion (RT-QuIC) assay

Recombinant PrP protein in 6M guanidine hydrochloride (GdnHCl) solution was diluted in RT-QuIC buffer (20 mM sodium phosphate pH 7.4; 130 mM NaCl; 10 mM EDTA; 0.002% SDS) to a final protein concentration of 0.2 mg/mL (and residual 0.2 M GdnHCl). Reactions were seeded with AF4 fractions containing the same amount of total PrP (based on immunoblotting intensity) in a final volume of 180 μL/well. The aggregation reactions were carried out in 96-well plates (white plate, clear bottom; Costar 3610) sealed with thermal adhesive film (08-408-240; Fisherbrand). The samples were incubated in the presence of 10 mM thioflavin T (ThT) at 42°C with cycles of 1 min shaking (700 rpm double orbital) and 1 min rest. ThT fluorescence measurements (450+/210 nm excitation and 480+/210 nm emission; bottom read) were collected every 60 minutes. There were three technical replicates per experiment.

## CAD5 cell infection assay

A murine catecholaminergic cell line lacking endogenous mouse PrP expression and stably transfected with hamster PrP, CAD5-PrP−/− (HaPrP) cells, were generously provided by Dr. Joel Watts of Tanz Centre for Research in Neurodegenerative Diseases, University of Toronto [37]. CAD5-PrP−/− (HaPrP) were grown in Opti-MEM medium (Thermo Fisher Scientific) containing 10% fetal bovine serum and 0.2% penicillin-streptomycin at 37°C with 5% CO2. Two milliliters of cells suspension were combined with 120 μL of AF4 fraction and incubated in a 6-well plate for two days. Then, the media was aspirated and the cells were mechanically passaged seven times every 2–3 days and harvested for immunoblotting analysis. Cells were homogenized in 50 μL ice cold RIPA buffer. Total protein concentration was determined by BCA assay. To detect PrP$^{res}$, 20 μg of protein (in 10 μL) were treated with 40 μg/mL Proteinase K (Invitrogen) at 37°C for 1 h. Then 2X loading buffer (187.5 mM Tris HCl pH 6.8, 15% glycerol, 15% SDS, 9 mM EDTA, 8 M urea, 8% βME) was added, samples were boiled for 10 min, and 15 μL were subjected to immunoblotting analysis using Sha31 (1:20,000 dilution) as primary antibody.

## Negative stain electron microscopy

Carbon-coated 200 mesh copper grids (Electron Microscopy Science, USA) were glow charged for 30 seconds using a Pelco Easy Glow 100 x glow discharge unit (Ted Pella Inc, USA). Microliter amounts (2–3 μl) of the purified fractions were adsorbed on the grids for up to 10 min by sitting drop method. The grids were washed three times (3 × 50 μL) with filtered ammonium acetate (0.1 M and 0.01 M), pH 7.4 and stained with filtered 2% (w/v) uranyl acetate (Electron Microscopy Science, USA) or filtered 2% (w/v) phosphotungstate (Sigma-Aldrich). Excess stain was removed using filter paper and the grids were air-dried. The samples were viewed with a Tecnai G20 transmission electron microscope (FEI Eindhoven, NL) using an acceleration voltage of 200 kV. Electron micrographs were recorded on an Eagle 4 k × 4 k CCD camera (FEI).

## Immunoblotting

Samples were prepared in Laemmli loading buffer containing SDS and 2-mercaptoethanol, boiled for 10 min, electrophoresed on 4–12% NuPAGE Bis-Tris precast gels using an Invitrogen system and transferred to polyvinyl difluoride (PVDF; Millipore) membranes (wet transfer). Blots were then incubated with primary 1:30,000 dilution of Sha31 antibodies (Spibio) in TBS-0.5% Tween 20 at 4°C overnight. 1:5,000 dilution of Sha31 was used for AF4 fractions. AP-tagged anti-mouse IgG (Promega) in TBS-0.1% Tween was used as secondary antibody. Blots were developed using AttoPhos substrate (Promega) and detected on an ImageQuant LAS 4000 (GE Healthcare). To compare distribution curves from different immunoblots, PrP

intensities from each fraction were first plotted against the average $R_h$ for that fraction, then the area under the curve for the entire run was calculated and normalized to 100% for each strain.

## Statistical considerations

One-way analysis of variance (ANOVA) was used to identify group-wise differences and post-hoc Tukey's test was used to identify pairwise differences ($p < 0.05$ considered significant). All analyses were performed using Graphpad Prism (v5).

## Supporting information

**S1 Fig.** Particle density (**A**) and UV signal at 280nm (**B**) for NBH (black), 263K (red), HY (green), and DY (blue) BHs fractionated by AF4. In fractions 46–50, higher particle density and UV signal is evident for prion-infected samples when compared with NBH.
(TIF)

**S2 Fig. Blank fractogram using 50 mM HEPES pH 7.4 containing 0.05% SDS and 50 mM (blue), or 150 mM (red) sodium chloride as running buffer.** SDS micelles present in the running buffer are evident in the DLS measurements, where background values are ~2–3 nm $R_h$ (red dots).
(TIF)

**S3 Fig. Size distributions of prion particles fractionated in the presence of SDS micelles are depicted as relative amount of PrP contributing to each particle size ($R_h$).** Fifty mM HEPES pH 7.4 containing 150 mM sodium chloride and 0.05% SDS was used as AF4 running buffer. Three brains for 263K (red, A-C), HY (green, D-F), and DY (blue, G-I) were analyzed.
(TIF)

**S1 Table. Particle density estimation for HY assuming a four-rung β-solenoid structure.** The $R_h$ of particles was calculated from DLS measurements. The length of the particles was calculated fitting the MALS measurements to a rod-shape model. The number of PrP monomers per PrP$^{Sc}$ particle was estimated from the PrP$^{Sc}$ particle length, assuming a contribution of 1.92 nm in length per PrP monomer and a PrP$^{Sc}$ fibril formed by two protofilaments. The PrP normalized intensity was determined by immunoblotting using Sha31 antibody. The relative number of PrP$^{Sc}$ particles was calculated by dividing the PrP normalized intensity by number of PrP monomers per PrP$^{Sc}$ particle. The maximum possible PrP$^{Sc}$ particle density was calculated with ASTRA software (see methods) for fractions 46 and 48, and extrapolated to the rest of the fractions based on the relative number of PrP$^{Sc}$ particles. (*) these values of particle density were calculated from MALS measurements directly, fitting the data to a rod-shape model, and assuming a rod radius of 5.5 nm. This calculation was performed for all samples; the specific values obtained for brain E are shown in the table.
(DOCX)

## Acknowledgments

We thank Dr. Michael Woodside and Craig Garen for providing the recombinant prion protein SHaPrP substrate used in the RT-QuIC experiments, Dr. Joel Watts for generously providing the CAD5-PrP–/– (HaPrP) cells and Wyatt Technology for personal assistance in the analysis of DLS and MALS data.

## Author Contributions

**Conceptualization:** Leonardo M. Cortez, Valerie L. Sim.

**Formal analysis:** Leonardo M. Cortez.

**Funding acquisition:** Leonardo M. Cortez, Debbie McKenzie, Valerie L. Sim.

**Investigation:** Leonardo M. Cortez, Satish K. Nemani, Camilo Duque Velásquez, Aishwarya Sriraman, YongLiang Wang.

**Methodology:** Leonardo M. Cortez, Satish K. Nemani.

**Project administration:** Leonardo M. Cortez, Valerie L. Sim.

**Resources:** Camilo Duque Velásquez, Holger Wille, Debbie McKenzie, Valerie L. Sim.

**Supervision:** Leonardo M. Cortez, Valerie L. Sim.

**Validation:** Holger Wille.

**Visualization:** Leonardo M. Cortez, Holger Wille.

**Writing – original draft:** Leonardo M. Cortez, Valerie L. Sim.

**Writing – review & editing:** Leonardo M. Cortez, Satish K. Nemani, Camilo Duque Velásquez, Holger Wille, Debbie McKenzie, Valerie L. Sim.

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
