## [Decision Letter · Decision Letter 0]

12 Mar 2021

Dear Dr Cortez,

Thank you very much for submitting your manuscript "Asymmetric-flow field-flow fractionation of prions reveals a strain-specific continuum of quaternary structures with a protease-resistant core forming at a hydrodynamic radius of 15 nm" for consideration at PLOS Pathogens. As with all papers reviewed by the journal, your manuscript was reviewed by members of the editorial board and by several independent reviewers. In light of the reviews (below this email), we would like to invite the resubmission of a significantly-revised version that takes into account the reviewers' comments.

We cannot make any decision about publication until we have seen the revised manuscript and your response to the reviewers' comments. Your revised manuscript is also likely to be sent to reviewers for further evaluation.

Sincerely,

Surachai Supattapone

Associate Editor

PLOS Pathogens

Neil Mabbott

Section Editor

PLOS Pathogens

Kasturi Haldar

Editor-in-Chief

PLOS Pathogens

orcid.org/0000-0001-5065-158X

Michael Malim

Editor-in-Chief

PLOS Pathogens

orcid.org/0000-0002-7699-2064

Reviewer's Responses to Questions

**Part I - Summary**

Reviewer #1: Cortez and colleagues present an intriguing comparison of three hamster scrapie strains based primarily on the AF4 technique combined with inline light scattering measurements. Their analyses provide valuable new information about the size range of PrPres particles of each strain in solubilized brain homogenates, and correlations of particle size to biochemical characteristics such as proteinase-K resistant and conformational stability. Also, as expected, the 263K and HY strains look similar, but not identical. On the other hand, the DY strain was quite distinct. Overall, the work is novel, thought-provoking and important. However, several issues could/should be considered to improve the manuscript.

Reviewer #2: The manuscript by Cortez et al analyzed the quaternary structure of PrPSc particles in three hamster prion strains by using the asymmetric-flow field-flow fractionation technique. This is probably the best technique to study the heterogeneity of protein aggregates in neurodegenerative diseases. Authors performed a detailed analysis of intra- and inter-strain heterogeneity of the PrPSc particles and revealed several differences among three prion strains. The most interesting finding is probably the observation that in all three prion strains, the PK-resistant PrP prominently accumulates in particles greater than 15 nm Rh.

Reviewer #3: Infectious prion protein (PrPSc) is composed of aggregates of a misfolded, pathogenic form of the host prion protein PrPC. For any given strain of prion, there is a heterogeneous mixture of PrPSc aggregates. However, it remains unclear exactly how this heterogeneity may contribute to different aspects of prion pathogenesis. The manuscript by Cortez et al. uses asymmetric-flow field flow fractionation (AF4) combined with dynamic and multi-angle light scattering to analyze the physical properties of aggregates from 3 hamster prion strains: 263K, Hyper (Hy) and Drowsy (Dy). They found a strong correlation of PrPSc aggregate structure with strain differences. They also found a strain-independent transition of PrPSc aggregate structure at a particle size of ~15nm, with no further evidence of structural change in larger particles. They conclude that PrPSc subpopulations that differ in their biochemical properties (e.g. stability, seeding activity) appear to define strain phenotypes.

This is a well-written, interesting, and thoughtful study which adds to our knowledge of how hamster prion strain phenotype may be influenced by PrPSc aggregate structure. The data are clearly presented, support the conclusions drawn, and would be of interest to prion researchers.

**Part II – Major Issues: Key Experiments Required for Acceptance**

Reviewer #1: 1) Most importantly, although the authors would have no way of knowing this prior to submission, the structure of the 263K PrPRes has recently been solved by cryo-EM and the structure is now posted on bioRXiv (https://doi.org/10.1101/2021.02.14.431014). This structure reveals that in contrast to the assumptions made in the current manuscript, the ordered fibril core is a single PIRIBS-based filament (13 x ~4 nm) and not a pair of 4-rung beta-solenoid-based protofilaments. This alters the fundamental assumptions about monomers per unit length that the authors have used for their calculations presented in Table 1 and elsewhere. Thus, these calculations should be redone accordingly. Although I doubt that such recalculations will materially alter the general conclusions of the manuscript, they will certainly make the estimates presented in Table 1 more accurate for 263K.

2) L206 and thereafter: In these calculations, it seems that the lateral bundling of fibrils, and its effects on mass per unit length calculations should also be considered. Measurements of the ratios of radii of hydration and gyration could give indication of how elongated the particles are and, hence, the extent to which the fibrils are bundled laterally.

Reviewer #2: 1. A major weakness is that the study did not directly address the relationship of various PrP aggregates to prion infectivity. It is understandable that animal bioassay is probably too costly, but authors could consider using cell based infectivity assay to at least show the correlation. Authors used RT-QuIC assay to show the seeding activity. So far, there is no evidence that RT-QuIC product is infectious and there is no study to definitely show a correlation between RT-QuIC positivity and prion infectivity. RT-QuIC seeding activity cannot reflect prion infectivity and pathogenicity.

2. The RT-QuIC assay showed seeding activity in 10 nm particles of all three strains, which is comparable to the seeding activity in larger particles. Since there is no PK-resistant PrP in these particles (at least for DY strain), some discussion about the possible seed in these fractions would help. This result itself is quite interesting if the same fraction can be tested for prion infectivity assay as mentioned above.

3. Could the solubilization step alter the size of PrP aggregates? Is it possible that the less stable DY strain is more vulnerable to the solubilization treatment, which results in more PrP being dissociated from the complex? In that case, various sizes of the particles may actually reflect the braking down products instead of native products in diseased brains.

4. The strong PK resistant signal in fractions greater than 15 nm Rh was interpreted as that “This result strongly suggests a strain independent conformational change in PrPSc structure once particles reach 15 nm Rh.” If the sizes of particles actually reflect the breaking down product during solubilization step, this observation could be interpreted in a different way, that the PK-resistant conformation cannot be maintained when the particle is smaller than 15 nm Rh.

Reviewer #3: None

**Part III – Minor Issues: Editorial and Data Presentation Modifications**

Reviewer #1: 3) L130-143: Another possible explanation for the lower ratio of PrPres:total PrP could be that DY has less PrPSc per unit of tissue, thus decreasing the PrPSc:PrPC ratio, and therefore the PrPres to total PrP ratio.

4) I cannot find reference to Fig. 2D in the main text.

5) L220: criteria should be criterion

6) L239: It would be helpful to refer to Fig. 2B here.

7) L275 and beyond: It is important to point out that this type of analysis only probes the PK sensitivity of the epitope of the antibody used, which is not clear here (to me at least). Other parts of the structure of a given strain may or may not be similarly PK-sensitive.

8) L284-5 (for example): The wording here (“conformational change in PrPSc structure”) and elsewhere implies that the authors assume that PK-sensitive PrP species that are bigger than PrPC in NBH, but smaller than 15 nm, are a form of PrPSc (which they define in the beginning as being infectious). What is the evidence that such species are a form of PrPSc, rather than, say, some non-infectious oligomeric intermediate that might be induced by PrPSc in the brain, but dissociated in detergent? In that case, a better description would be that PrPSc particles that are stable under these experimental conditions have an Rh of > ~15 nm.

9) L289: Analogous to (7) above, this analysis probes only the stability of the epitope of the antibody used, not the whole particle, and should be described as such.

10) RT-QuIC data, Fig 7: It would be informative to also show the relative seeding activities as a function of size before they are adjusted for PrP concentration. I assume, based on Fig 2C, that roughly 4x more equivalents of the 10 nm fraction would have been used to seed the reactions than the larger particle fractions. Such results might weigh in on the prior question of whether such particles are really some form of PrPSc. The existence of such particles with seeding activity would be very interesting in their own right, but not if the seeding activity in the 10 nm fraction is simply due to spillover of larger seeds from adjacent fractions during fractionation. In any case, such quantitative comparisons are difficult to make accurately based on RT-QuIC lag phase alone.

10) L315-316: A thought to consider: This result would be the expected result if HY and 263k were largely single fibrils for which only the ends have seeding activity. In this case large fibrils have the same number of templating surfaces (i.e., 2) as smaller ones. On the other hand, the larger DY particles may either be non-fibrillar assemblies with more than 2 templating surfaces per particle, or bundles of shorter fibrils such that the bigger the particle, the more templating surfaces it has per particle.

11) L367: Again: how does one know that the PrPsen components have any replication activity (or infectivity?

12) L434: I think that "the main contributor" is too strong here because this work has not probed or defined secondary or tertiary structures, and, therefore, these structures cannot be assumed to be the same for these strains. In fact, FTIR comparison of these strains provide evidence that the beta sheet secondary structure of DY is significantly different from HY and 263K (ref 40). Thus, "a significant contributor" would be more appropriate here and elsewhere when this conclusion is expressed.

Reviewer #2: Since PrPSc generally means the PK-resistant and infectious PrP form, the term PrPSc used in this manuscript should be defined.

Reviewer #3: 1) Are there any EM data showing the presence of fibrillar structures in the different AF4 fractions, particularly those above 15 nm? These data would provide further justification for fitting the light scattering data to a rod-shaped model (lines 378-380). It would also provide support for assuming that fibrillar assemblies are present in fractions which contain larger PrPSc particles.

2) Light scattering data is used to fit PrPSc particles to a rod-shaped model assuming 2-protofilament fibrils and a beta-solenoid model of PrPSc structure (lines 204-209, 378-380). A model based on a parallel in-register intermolecular beta sheet (PIRIBS) structure has also been proposed for hamster 263K PrPSc [J Biol Chem 289: 24129 (2014)]. Presumably, fitting light scattering data to the PIRIBS model would lead to different particle sizes which may or may not impact some of the conclusions of the manuscript. The authors might consider briefly addressing their data in the context of the PIRIBS model and whether or not it impacts the major conclusions of their study.

3) For the data in Figure 7, can the authors calculate seeding activity/particle using the available data? That would allow a more direct comparison between seeding activity and particle size for 263K, Hy, and Dy hamster prions.

4) In the Conclusions (lines 434-444), the authors state that their results show that the proportion of distinctive PrPSc subpopulations determines the prion strain phenotype. In the absence of direct experimental confirmation that this is the case, “determines” is too strong a word in this context. While they can certainly draw a correlation between the properties of distinctive PrPSc populations and prion disease incubation time, the data do not support a determinative role for a specific population of PrPSc quaternary structure in dictating prion strain phenotype. The authors should consider modifying this statement.

5) The text on lines 201-203 refers to the PrPSc particle densities for fractions 46-50, lists what appears to be an average particle size plus and minus some undefined error with no units given, and then references Table 1. However, these data are not in Table 1. The table shows calculations for particle densities for a single brain and does not even contain particle densities for fractions 49 and 50. The authors need to be more precise about what they are discussing in this section.

6) Do lines 214-218 refer to Figure 2D? Please clarify.

7) In Figure 3, there are no data with a significance indicated by * or ** as described in the legend. Please fix to make sure that the legend and figure match.

8) Reference 20 needs to be updated to show the complete reference for the final, published manuscript.

**Do you want your identity to be public for this peer review?** 

Reviewer #1: **Yes: **Byron Caughey

Reviewer #2: No

Reviewer #3: No
---

## [Decision Letter · Decision Letter 1]

9 Jun 2021

Dear Dr Cortez,

We are pleased to inform you that your manuscript 'Asymmetric-flow field-flow fractionation of prions reveals a strain-specific continuum of quaternary structures with protease resistance developing at a hydrodynamic radius of 15 nm' has been provisionally accepted for publication in PLOS Pathogens.

Best regards,

Surachai Supattapone

Associate Editor

PLOS Pathogens

Neil Mabbott

Section Editor

PLOS Pathogens

Kasturi Haldar

Editor-in-Chief

PLOS Pathogens

orcid.org/0000-0001-5065-158X

Michael Malim

Editor-in-Chief

PLOS Pathogens

orcid.org/0000-0002-7699-2064

Reviewer Comments (if any, and for reference):

Reviewer's Responses to Questions

**Part I - Summary**

Reviewer #1: The authors have addressed my concerns adequately.

Reviewer #2: The revised manuscript properly addressed the concerns raised by me and other reviewers during the first round of review. Authors' response is excellent and very responsive.

Reviewer #3: The new data significantly improve the paper and strengthen the main conclusions. The revised manuscript addresses all of my concerns with the original manuscript.

**Part II – Major Issues: Key Experiments Required for Acceptance**

Reviewer #1: (No Response)

Reviewer #2: None

Reviewer #3: None

**Part III – Minor Issues: Editorial and Data Presentation Modifications**

Reviewer #1: (No Response)

Reviewer #2: None

Reviewer #3: The authors addressed all of my original concerns.

PLOS authors have the option to publish the peer review history of their article (what does this mean?). If published, this will include your full peer review and any attached files.

Reviewer #1: No

Reviewer #2: No

Reviewer #3: No

---

## [Editor Report · Acceptance letter]

22 Jun 2021

Dear Dr Cortez,

We are delighted to inform you that your manuscript, "Asymmetric-flow field-flow fractionation of prions reveals a strain-specific continuum of quaternary structures with protease resistance developing at a hydrodynamic radius of 15 nm," has been formally accepted for publication in PLOS Pathogens.

Best regards,

Kasturi Haldar

Editor-in-Chief

PLOS Pathogens

orcid.org/0000-0001-5065-158X

Michael Malim

Editor-in-Chief

PLOS Pathogens

orcid.org/0000-0002-7699-2064